# Sustainable Leadership: Philosophical and Practical Approach in Organizations

**Vasile-Petru Hategan** [1] and **Camelia-Daniela Hategan** [2,*]

1 Institute of Media and Social-Humanitarian Sciences, South Ural State University,
454080 Chelyabinsk, Russia; vphategan@gmail.com

2 Department of Accounting and Audit, Faculty of Economics and Business Administration,
West University of Timisoara, 16 Pestalozzi Street, 300115 Timisoara, Romania

* Correspondence: camelia.hategan@e-uvt.ro

**Abstract:** The contemporary leadership concept manifests trends of continuous improvement, which is why this paper is an approach to the field from the perspective of philosophical counseling practices, different from current training and development programs for managers. Thus, the new opportunities are identified, which can join business education and personal development programs for leaders. The paper analyzes the characteristics of coaching studies compared to those of philosophical counseling applied in organizations, using the concept of philosophical leadership in order to identify the existence of interdisciplinary links between the two fields. The intensity of these links was studied through the bibliometric analysis performed on the concepts of leadership and philosophy, which were the subject of papers published in relevant journals indexed in the international database Clarivate Analytics–Web of Science Core Collection. The results of the analysis showed a close connection between concepts, which confirms their association. Through the practical applications of philosophy in the leadership, an innovation of the programs takes place of current specialization dedicated to leaders, which contain tools and methods that can be taken from philosophical practices, to achieve high-performance leadership programs. The concept of philosophical leadership can have a sustainable approach which can be included in personal development programs addressed to leaders, and will be delivered in the form of leader service packages, in which other specialists can participate, such as philosophical counseling, who can be successfully involved in the development of the leadership training program (LTP).

**Keywords:** sustainable leadership; organization; philosophical leadership; coaching for managers; leadership training program (LTP); business education

## 1. Introduction

With the rediscovery of philosophical practices in Europe and around the world, and the establishment of the first associations of practitioners, some philosophers moved from the academic to the applied area, by opening philosophical practice offices and offering counseling services, which later led to the emergence of the first specialists in philosophical and ethical counseling. Later, the area of specialization diversified through the emergence of other specializations, such as bioethicists, group facilitators or organizational philosophical consultants, who migrated from the area of philosophy or other fields, through training or education programs managed by universities or professional associations. We notice the tendency of practitioners to organize regular meetings (every two years) for exchanges of information, presentations of works or results derived from practice, applied seminars and workshops for training. All this information has been collected and published in collective volumes of papers [1–9], which has reduced the interest of practitioners to publish papers in scientific journals; this concern is sporadically manifested only by those who have kept in touch with the academia or those engaged in research projects conducted in the new field of applied philosophy [10].

The promotion of philosophical practice was mainly achieved through author volumes or the collective ones already mentioned, and only occasionally were articles published in professional journals or other scientific publications. We propose in the paper to conduct a bibliometric analysis in the literature, especially papers published in journals indexed in the Web of Science (WoS), to highlight the current situation and show the still low interest in this type of promotion of information on practical applications of philosophy, in the form of specialized counseling and consultancy. The beneficiaries of this study are those interested in conducting university projects and specialization and training programs, or the specialists involved in continuous professional training and the professional associations interested in developing training, specialization and improvement programs in the field. Practitioners and organizational consultants are the first recipients who can be trained in a specialization program, and who at the end of the training can offer their services to companies and institutions interested in improving and developing a successful leadership. To have this analysis and the proposed purpose, in the debate section we present the resulting arguments, to give an answer to the following questions:

Q1: What contribution does philosophical practice have to the development of leadership?
Q2: Are there interdisciplinary connections of philosophical practices with other fields involved in leader development?
Q3: Can the philosophical trend be included in leadership training?

This study is an exploration due to the rarity of the previous literature, but which leads us to frame the theme in the definition of innovation, this being materialized through sustained specialization, which introduces concepts and tools specific to philosophical practice in the leadership process. The conclusions of the paper will outline new perspectives for collaboration between various fields, through the trends manifested in the sphere of organizational philosophical counseling, which can be applied for the common benefit of the leader and organization. The purpose of taking over these philosophical practices is highlighted by presenting the concept of philosophical leadership, supported to be implemented in all leadership training programs, with the help of specialists in organizational philosophical consulting who can be attracted by this process.

The paper is organized as follows: Section 2 presents the theoretical background, Section 3 describes the research methodology, and Section 4 presents the results of the comparative study on concepts chosen to be studied and of the bibliometric analysis regarding the fields analyzed in the paper. The paper is completed with Section 5 in which the results obtained and future directions of research are discussed, followed by Section 6 in which the conclusions are presented.

## 2. Theoretical Background

To illustrate the potential of applying philosophy in the business environment, we start from the premise that philosophy has also entered other fields, through its interdisciplinary links that can be developed, one of them being the economic field. The presentation of the business is often conducted using the expression "the philosophy of the company or organization", a statement by which the organization defines its mission, indicating its own way of achieving the proposed goal. Thus, appeared the business consultant, a specialist who practices counseling for managing a company, who uses techniques and methods specific to managerial leadership, and who can also use tools taken from philosophy. As an alternative to this practice, we have identified some philosophical approaches in the training and development of managers, using tools specific to philosophical practices, where reflective practice and Socratic dialog can be successfully applied within the organization.

The process brings an improvement in the training of managers, called the "philosophy of management" [11] or "philosophy of labor" when referring to other workers within the organization [12]. We find that the services of a philosophical practitioner can be intended for managers, management teams or groups of workers in the organization. In investigating the need to introduce philosophy into the business environment, managers were identified, who are interested in collaborating with philosophical practitioners and who are only

interested in solving ethical problems, or managers with philosophy studies who support building a professional reputation of the philosophical advisor role in the organizational and business field. Business ethics are presented by Hoogendijk as a tool that improves the relationship between the management, company staff, society and the environment, being one of the first philosophical practitioners dedicated to advising companies [13,14]. It differentiates between business and philosophy to show what a philosopher can perform and connects philosophy with management, discussing the work of a philosopher who runs Philips. The approach was followed by other practitioners who brought philosophical tools in their practices within a company or organization [15], other authors presented the experience gained in the field in a volume in which they transformed the ancient concept of school into the concept called "space free from reflection" [16].

Practitioners are the ones who noticed the difference between philosophical counseling and other types of counseling, showing that philosophy has a healthy discourse that can be considered essential for developing the company. Nokia collaborated with a philosopher, Esa Saarinen, who introduced the concept of "space for reflection," summarizing his mission as an advisor by "helping people flourish" [16]. Inspired by his work, another professor of philosophy, Tom Morris, published two successful books (*If Aristotle ran General Motors* and *If Harry Potter ran General Electric*), offering suggestions and topics for thinking in the organizational field [17,18]. In Europe, with the appearance of the first philosophical practice cabinet, opened in 1981 in Germany by the philosopher Gerd Achenbach, it was the signal followed by other professionals who left the university chair to move to the university chair of applied philosophy. The French philosopher Vegleris is one of those who developed conferences and seminars, and created consulting services for businesses and managers [19]. Vegleris supported the involvement of philosophical practitioners in the business environment, by providing managers with tools for achieving results or a better life, showing that philosophy produces clarity of thought, generates confrontations of ideas and gives meaning to the reality that a manager faces, by learning the art of asking questions and reflecting on life issues [20].

The University of Lucerne was the initiator of a course in applied philosophy for managers, organized by Professor Enno, to bring philosophy closer to the Swiss entrepreneur [21]. Although philosophy is often perceived in the business environment as a waste of time, there are philosophers who have been concerned in supporting entrepreneurs, by emphasizing the importance of the worldview, with a recommendation for all those starting a business to define their vision calling on the help of a specialist, who can also be the philosophical advisor [22]. Concerns regarding the philosophy applied to companies were also outside philosophy through Andrea Vitullo, who rhetorically asked herself: "What would Socrates do today?", saying that philosophy can penetrate among current managers or boards, bringing to attention a new concept, of reflective leadership [23], where Vitullo analyzes the characteristics of a leader, starting from the emotions he can be subjected to, highlighting coaching techniques that can be helpful, and proposes the transformation of the leader with the help of practical philosophy, and by using the Socratic dialog, it is possible to transition from leader to person [24,25]. Thus, the concept of philosophical leadership [26] (Marino 2008) was developed by practitioners specializing in philosophical practices for leaders, to give meaning to a different leadership with the help of the tools offered by philosophy, which are made available to an organization through the services of a philosophical advisor. References that can be assimilated into the current concept of leadership have been noticed by researchers who have studied this aspect in the works of important philosophers, such as Aristotle, Locke, Rousseau, Hegel, Marx, Nietzsche [27], or others, such as Epicurus, Seneca, Heraclitus, Popper, Leibniz, Bacon, Betham, Ockham, Russel, Whitehead, Pascal or Kant [28].

Other authors draw attention to Plato's Republic to highlight the role that the wisdom of humanity can play in the preparation and practice of contemporary leadership [29] or highlight philosophical ethical tools from the thinking of philosophers such as Foucault, Kant and Butler, which can be included in leadership development programs [30]. To

answer the question of the areas in which a philosophical advisor can work in a company, two Italian practitioners present the answer in the form of a "catalog of philosophical-practical products" addressed to organizations, offering services: for people and groups, problem-oriented services, and professional development and training services for the organization's staff [31]. Within an organization, Socratic dialog as a working tool for groups or individuals can be successfully applied to analyze and deepen the understanding of concepts, to build a consensus on a dilemma or problem, to support communication and dialog between participants, or to increase capacity reflection in a group.

Among the practitioners on the North American continent, the philosopher Peter Koestenbaum has been noted since the 1970s, concerned with the role of philosophy in the person [32], later defined as the corporate philosopher, for his experience in the organizational field [33] proposing a model of practice called Leadership Diamond, in which it presents the elements for achieving the excellence of the leader [34]. The model presents four strategies used for this purpose, to which the author assigns a tactic, namely: to achieve the vision the tactic is to think innovatively; for the discovery of reality the tactic is to give up illusions; the ethical approach is made with the help of services; courage has, as a tactic, the promotion of the initiative of the person advised by the practitioner [34]. A pioneer of US philosophical practice is the philosopher Marinoff, who, since 2002, referred to the first attempts of practitioners to enter the organizational environment, through practitioners specializing in drafting mission statements and drafting ethical codes necessary for a company that has passed to the presentation of seminars on their implementation, which made the company liable to be mitigated in the event of damage to its own staff [35]. In the guide to philosophical practice [36] developed by Marinoff, the forms of philosophical practices for organizations are presented, namely: the motivational discourse given in corporate events; drafting a code of ethics of the organization, and organizing workshops for its implementation; ethical compliance, achieved through the implementation of ethical norms; moral self-defense, manifested through philosophical workshops of the ethical clarification of a problem; short Socratic dialog, adapted to the corporate time requirements given to philosophical counselors; dilemma training, which applied the concept of putting in difficulty, used to solve work problems; as well as the methods that define it, referred to by acronyms as PEACE and MEANS and which are applied in organizations [37,38].

We notice that in the organizational field, the first to start the elaboration of codes of ethics were the business advisors, followed by the philosophical practitioners who were concerned at the beginning of the activity with the implementation of ethical norms in American companies. Thus, in the organizational framework was agreed a short form of Socratic dialogue, which requires more facilitators by actively intervening in finding a point agreed by all participants, and about this tool Marinoff states that "to experience briefly Socratic dialogue is preferable to omitting it" [36] (p. 177). The training dilemma is a technique that initially emerged to manage the dilemmas that would arise in a conflict between morality and ethics. In the 1980s, a certain systematization was conducted by Henk van Luijk, a professor of business ethics, who worked on a research project at the European Institute for Business Ethics. This practice is described by Marinoff in his philosophical guide, in the form of work steps, as follows: announcing a problem generated by a conflict, followed by identifying participants and some data about their conflict, establishing the dilemma and whose task it is, followed by arguments for and against the dilemma under debate and at the end, there is a decision, which can continue with the justification of the decision taken [36] (p. 174). The contribution of philosophy to the development of contemporary leadership is made in the organizational environment, through philosophical practitioners who can become either consultants, coaches, trainers or facilitators, who use philosophical tools, most often used in the organizational environment being the Socratic dialog [39] that revives tradition Socratic and combats managerial automatisms by activating the thinking and judgment skills necessary for a leader.

Other practitioners have performed the activity of organizational coaching with the help of philosophical approaches, understanding that it can become philosophical counseling [40] through the multitude of meanings that philosophical practice can offer. This service can be valued through the concepts and tools provided by philosophy, the trend being considered as a real innovation in the field of services for organizations. Leadership can, thus, receive a personal meaning related to the human being, to the detriment of the initial goal pursued in training a manager, focused on obtaining an economic result (profit), the new approach will place the organization within the community, becoming more socially and community responsible, with the help of the philosophy applied in organizations through the philosophical practice services promoted by the new specialists.

Another concept, of philosophical sensitivity, was developed in Italy [41], and authors suggested the use of philosophical exercises based on the art of asking questions, stating that a well-formulated question will generate a constructive dialogue, which shows the need for the development of some dialogic competencies of the leader, following the model promoted by Socrates in the ancient Agora, and transposed into the contemporary agora now identified with the organization [23] (p. 119). Returning to the corporate philosopher Koestenbaum, he was concerned since the 1980s with attracting philosophy in the development of leadership; he studied the concept of 'the authentic leader', identifying the essential criteria that define it, such as: motivation, experience, self-discipline, action, respect, authenticity, creativity, vision, manifestation of power, effectiveness, wisdom and morality [42].

In the same organizational context, new competencies necessary for the training of specialists in counseling and specialized consulting have been defined, all based on concepts and tools specific to philosophy and which are valued in the organizational environment with the help of philosophy practitioners [43]. Leadership can become a philosophy in action, through the practices offered by specialists called facilitators or philosophical organizational consultants, who can support through their services the leader to build their own vision of the world and life, which implicitly influences professional performance and personal development and evolution the leader. The main philosophical skills offered by philosophical counseling are manifested in several areas of development, such as emancipation, empathy, imagination, transformation, and leadership efficiency; the latter area can become "the philosophical practice of thinking" [44] (p. 157). Thus, the leader can define more clearly the direction in which he can navigate professionally, and not only, after discovering himself, with the help of the innovative concept, defined by the concept of philosophical leadership which we analyze in the paper. The conclusion regarding the application of philosophical counseling in the organizational field is given by Marinoff, as follows: "The philosophical consultant helps organizations be more virtuous. It is the highest possible calling for a philosopher and the greatest aspiration for an organization" [36] (p. 171).

## 3. Research Methodology

In order to initiate the study, we performed a comparative analysis of some elements characteristic of the practices used in the leadership formation process, starting from the applied personal development and coaching programs, and attracting in the leadership process the programs initiated and developed by the philosophical practice philosophical leadership applied in organizations.

In this research, we aimed to perform a bibliometric analysis, to identify trends and approaches in the field of applied organizational philosophy and leadership, and to study which authors have had an impact in the field analyzed in this way. We used VOSviewer software to identify the relationships between keywords used in the literature and between authors and their citations. the bibliometric analysis was limited to searching the Web of Science (WoS)—Clarivate Analytics database—, for specific topics of the current research topic, such as "philosophy" and "leadership", related to the period 1979–2021. The database highlighted a number of 1259 papers from all types of documents. The list was saved as

a .txt file and a thesaurus file was provided to combine the almost identical terms. This file was subsequently processed via VOSviewer software, providing keyword analysis and citations based on individual authors and country-by-country dispersion of papers, made after the authors' affiliation.

## 4. Results

### 4.1. Comparative Analysis of Two Personal Development Programs Used in Leadership

The leader usually comes from the staff dedicated to the management of an organization or entity, being trained in a leadership process carried out through a leader development and training program; such a representative program being the coaching service. We present in the comparative table some of the characteristics and basic elements of the two programs that can be used in the formation and development of the leader, respectively, coaching versus philosophical consulting.

In Table 1 we observe some differences generated by the approach of each program, but also the common interests of the two types of analyzed programs, manifested by the form of services offered to people and organizations interested in the development of the leader. In our opinion, these services can be offered in the form of a coaching program or personal development practice, which can be completed with a program called philosophical leadership, both of which are intended for the manager or leader of an organization. We support the opinion already expressed, that the practitioners involved in these forms of training leaders, regardless of their specialization, in collaboration to achieve a common package of personal development and leadership services [45,46]. The characteristics presented in Table 2 were identified by studying several sources [23,40,47] that were analyzed with those identified for organizational-applied philosophical practices [23,26] that can develop a new type of leadership based on humanistic consultancy [48]. We believe in the potential of the two directions of action, in which coaching can have philosophical approaches or philosophical practices that can borrow elements or strategies specific to organizational coaching, which can be successfully activated [45,47], including times of crisis [49].

**Table 1.** Comparative elements of philosophical practice and coaching.

| Comparative Elements | Coaching/Personal Development | Consultancy/Philosophical Leadership |
| --- | --- | --- |
| The context of the activity | Occupation is recognized | It is under a regulation process |
| The purpose of the action | The coach follows the development of the trained person | The counselor's concern is to clarify some situations or solve some dilemmas |
| Kind of action | As a result, it aims to reach a goal | It does not aim to achieve a specific result |
| The purpose of the action | Preparing the manager for professional success | Support the person in identifying their own vision of life |
| Procedure | Using a specific coaching method, establishing a work plan and individualization | Based on the counseling requirement, no method is applied, the application is adapted to the client's needs |
| The relationship between specialist/client | Collaboration | To facilitate counseling |
| Approach | Supports the person in achieving professional goals | It starts from the problem/dilemma and touches on essential issues of the leader's life |
| Operator training | Personal development coach/specialist | Philosophers, practitioners/philosophical counseling specialists, |
| Operator concerns | Development/transformation of the person using development/educational programs | Facilitating the solution of some problems/dilemmas of the advised person |
| Assessments on the person | With evaluations on the development of the person's professional career | No evaluations are made, but the development of the person is stimulated |
| The meaning of the action | Generating benefits for the person and building the successful strategy | Creating one's own vision with a role in leadership development |

**Table 1.** *Cont.*

| Comparative Elements | Coaching/Personal Development | Consultancy/Philosophical Leadership |
| --- | --- | --- |
| Objectives pursued | Develops a process of learning and changing attitude, developing leadership skills | Develop dialogue to solve dilemmas, using philosophical tools |
| How activate are the person's resources | By developing managerial and leadership skills | By stimulating leadership and solving dilemmas |
| Operating requirements | The process cannot be terminated unilaterally, has the duration of application defined in the contract, and can create dependence on the coach | The process can be stopped unilaterally, it takes place in relatively short periods of time, without generating dependencies on the specialist. |
| Type of service provided | Contract-based service | Fee payment |

Source: own processing.

**Table 2.** The main elements identified within the clusters.

| Cluster 1 (Red) | Cluster 2 (Blue) | Cluster 3 (Violet) | Cluster 4 (Green) |
| --- | --- | --- | --- |
| **Philosophy** | **Leadership** | **Attitudes** | **Management** |
| organization | education | perspectives | performance |
| model | health | identity | culture |
| kethics | knowledge | gender | implementation |
| business | perspective | spirituality | work |
| transformational leadership | strategy | sustainability | impact |
| values | care | diversity | quality |
| behavior | students | perceptions | framework |
| ethical leadership | innovation | challenges | lessons |
| CSR | success | communication | systems |

Source: the information was summarized by author from analysis report.

### 4.2. Bibliometric Analysis on the Concepts of Leadership and Philosophy

The bibliometric analysis performed started from the basic keywords specific to two analyzed fields, namely, leadership and philosophy, which were identified in the WoS, which resulted in a number of 1259 papers, for the analyzed period from 1979 to 2021. During this period, identified by the publication of the first paper in 1979, it had different evolutions, so that in the first 21 years only 100 papers were published (8%), and the next 20 years had an exponential increase to 1130 of papers (90%), the current year registering at the date of this study a number of 29 published papers (2%). We highlight, in Figure 1, the evolution of papers published in last 25 years from the analyzed period, which contain the topics introduced in the bibliometric analysis.

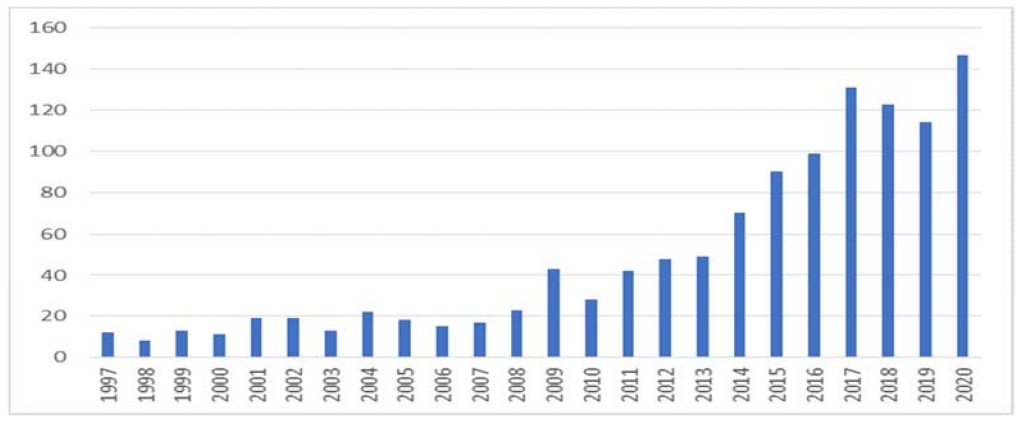

**Figure 1.** Number of papers published by years. Source: generated by the WoS analysis report.

Figure 1 shows that in the first decade of the analyzed period, the number of papers did not exceed 20 papers per year, after which the trend increased, reaching the end of the period of around 100 papers per year. The last four years of the period show an upward trend of the topic, so that in 2020 the number reached 147 published papers. From the viewpoint of the document type, the following weights were identified in the analysis sample: articles 75%, proceedings papers 15%, book chapters 5%, reviews 4%, and other types 1%. The domains of papers published by the authors were diverse, starting from the basic domains that define the concept of leadership, respectively, business and management, but also fields that can be considered by the border with which they can interfere, such as ethics, economics, social sciences, environment, finance, public administration or political sciences. Figure 2 shows the first 25 WoS categories, in which the journals were included, which contain papers on the analyzed topic.

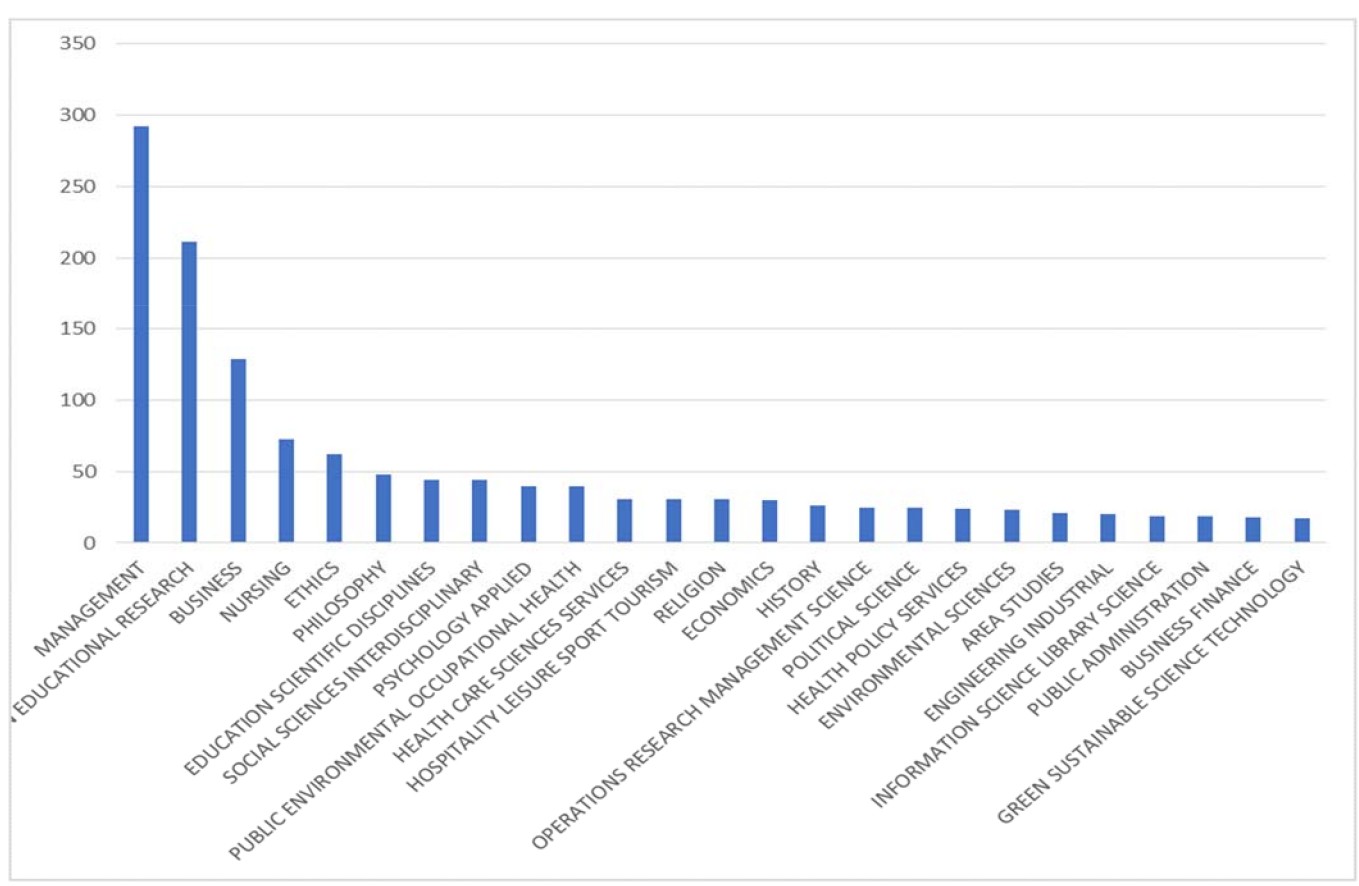

**Figure 2.** Distribution on WoS categories of the analyzed papers. Source: generated by the Web of Science analysis report.

From Figure 2, we observe a greater concentration of papers published in categories already dedicated to the studied field, such as: Management, Educational Research and Business, followed by other adjacent fields, respectively: Ethics, Philosophy and Social Sciences Interdisciplinary, which placed on the following positions in the chart. The other categories of journals presented in the figure show that other fields are also interested in publishing papers on leadership, which highlights other interdisciplinary links that can be studied.

4.2.1. Analysis by Keywords

In this section, the keyword analysis is presented, respectively, in the way in which most common keywords appeared together in the studied papers. The VOSviewer software highlighted 4634 keywords. From the total number of words selected, we took into analysis

those that had at least 10 occurrence, resulting in a portfolio of 85 words, from which 4 irrelevant words were removed (e.g., Ubuntu, context) and 6 words were replaced with similar expressions (e.g., business, organizations, strategies). Thus, 75 words grouped into four clusters remained in the final analysis, with a min of 10 words per cluster, having a setting made in the bibliometric software for resolution one. These cluster groups are detailed in Figure 3, where they are marked with different color.

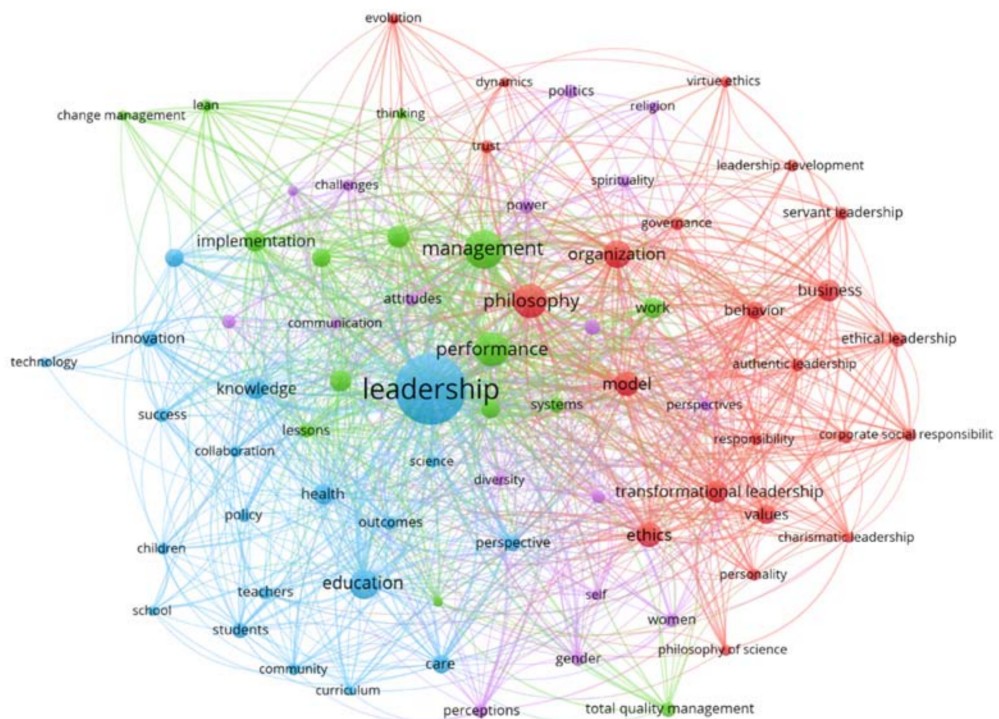

**Figure 3.** Keywords visualized. Source: computed in VOSviewer by the author.

Figure 3 created in VOSviewer has the following structure: the larger dots represent the most frequently mentioned keywords, the thicker lines represent how often the keywords are found together, and the distance between the dots represents the strength of the keyword relationship. The central cluster, colored blue, shows the central positioning of leadership, which made connections with three other clusters, respectively: management/performance, colored green; philosophy/ethics, colored red; attitudes, colored violet. We see in Figure 3 that the four identified clusters have different weights, being grouped two by two, where leadership and philosophy hold the majority and central positions with similar sizes, these being connected with two other smaller clusters, respectively, those representing management and attitudes, and which are positioned between the two majorities. We highlight in Table 2 the main elements of each cluster represented in the analysis, being identified in the network from the previous figure, and which actively interacted with each other.

Table 2 shows that the leadership cluster includes important elements that define it, such as education, knowledge, strategy and perspective, success, and, last but not least, innovation, which confirms the chosen research topic as being an important trend in the contemporary evolution of leadership. Another cluster resulting from the analysis was that of philosophy, which also contains elements derived from its connection with the field of leadership; we refer to a mixed concept, such as transformational leadership and ethical leadership with the most links, followed by the expressions leadership development, authentic leadership, servant leadership, charismatic leadership. Along with these elements, the analysis highlighted other important concepts, such as business, organization, ethics, values, model, behavior or responsibility, all indicating the interdisciplinary links that can

be developed by philosophy through its practices. The other two clusters identified in the analysis each generated elements characteristic of the concept marked to be defining, and emphasize the importance of the choice made in defining the analyzed domain. In the attitudes cluster we found grouped words that reflect personal aspects, such as perspectives, perceptions, challenges and communication. Sustainability has a high occurrence in this cluster, which shows that it has an important link in developing leadership by changing attitudes. For the management cluster, we identify elements characteristic of the field, such as performance, implementation, system, work or lessons, with tendencies towards impact and quality.

### 4.2.2. Citations by Authors' Analysis

The analysis presents the most cited authors who have published papers on leadership and philosophy that are indexed in the WoS database. Using the threshold settings of at least two documents per author (which were the default) and at least five citations, VOSviewer highlighted 59 authors who met these conditions. From the data processing, it was found that there are no close relationships between authors in terms of co-authorship and citation, the resulting graphic representation being irrelevant. Table 3 presents a top 10 of the authors with the most citations. Thus, the analysis showed that the most cited authors were as follows: Tsui with 195 citations from two documents, Luo with 171 citations from three documents, followed by Simpson with 85 citations from four documents. The maximum number of papers by an author (from those included in the tables) is five, written by the author Frunză, who has 30 citations, being located only on the tenth position in the top presented.

**Table 3.** Top 10 authors by citations.

| Author | Documents | Citations | Reference of Cited Paper |
|---|:---:|:---:|:---:|
| Tsui, Anne S. | 2 | 195 | [50] |
| Luo, Yadong | 3 | 171 | [50] |
| Simpson, Peter | 4 | 85 | [51,52] |
| French, Robert | 3 | 51 | [52] |
| Tomkins, Leah | 3 | 47 | [51] |
| Wilderom, Celeste P. | 2 | 44 | [53] |
| Xing, Yijun | 2 | 40 | [54] |
| Aij, Kjeld Harald | 2 | 38 | [55] |
| Wilkinson, Jane | 3 | 31 | [56] |
| Frunza, Sandu | 5 | 30 | [57] |

Source: The information is summarized by author from analysis report.

In the analysis, we proceeded to the distribution by countries regarding the affiliation of the authors, identifying some 97 countries, of which the first 5 countries hold two thirds of this total, respectively: USA (34%), England (13%), Australia (9%), Canada (7%) and China (5%). Figure 4 shows the countries with authors who have published more than 20 papers (16 countries).

In Figure 4, we can see that the representation on continents is either singular, with a single country, (e.g., Australia), or cumulated by component countries, except for the South American continent which has no representation in the figure. Even with this distributions of continents, no links between countries were identified, their graphical representation being irrelevant.

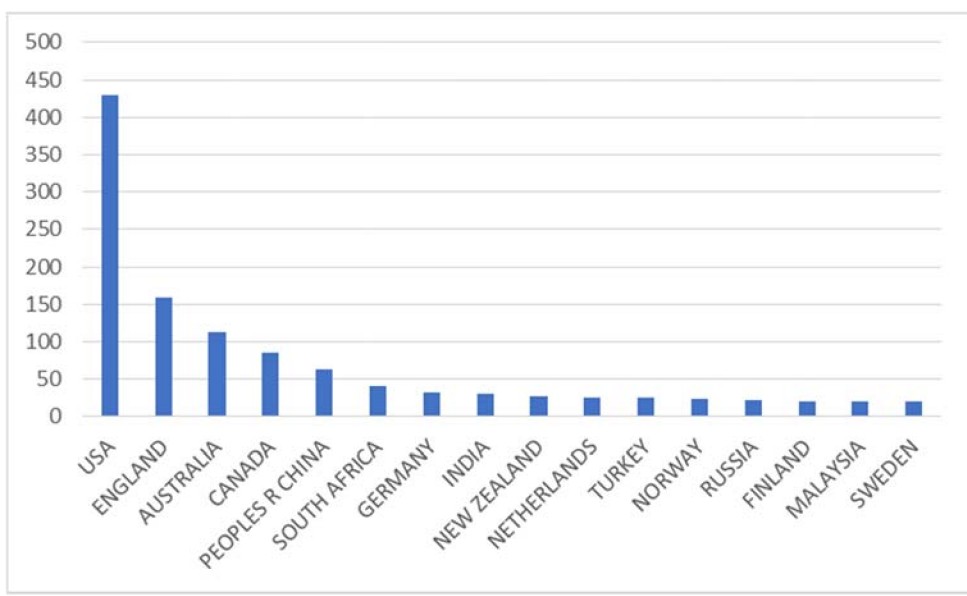

**Figure 4.** Distribution by countries of the analyzed papers. Source: generated by the Web of Science analysis report.

4.2.3. Journals Analysis

The papers analyzed were published in prestigious journals, being indexed in WoS by different categories (Figure 2), and from which we selected a top number of journals in which at least five papers on the topic were published, the situation being presented in Table 4.

**Table 4.** The journals that published on the topic.

| Name of Publications | Number of Papers | Publisher | WoS Core Collection * |
|---|---|---|---|
| *Journal of Business Ethics* | 30 | Springer | SSCI |
| *Total Quality Management & Business Excellence* | 12 | Routledge Journals, Taylor & Francis Ltd. | SSCI |
| *Philosophy of Management* | 10 | Springer International Publishing Ag | ESCI |
| *Chinese Management Studies* | 8 | Emerald Group Publishing Ltd. | SSCI |
| *Leadership Quarterly* | 8 | Elsevier Science Inc. | SSCI |
| *Educational Administration Quarterly* | 7 | Sage Publications Inc. | SSCI |
| *Leadership* | 7 | Sage Publications Inc. | SSCI |
| *Business Ethics Quarterly* | 6 | Cambridge Univ. Press | SSCI |
| *Cross Cultural & Strategic Management* | 6 | Emerald Group Publishing Ltd. | SSCI |
| *Human Relations* | 6 | Sage Publications Ltd. | SSCI |
| *Academic Medicine* | 5 | Lippincott Williams & Wilkins | SCIE |
| *BMC Health Services Research* | 5 | BMC | SCIE |
| *Educational Philosophy and Theory* | 5 | Taylor & Francis Ltd. | SSCI |
| *Journal of Management Inquiry* | 5 | Sage Publications Inc. | SSCI |
| *Journal of Organizational Change Management* | 5 | Emerald Group Publishing Ltd. | SSCI |
| *Sa Journal of Human Resource Management* | 5 | AOSIS | ESCI |
| *Sustainability* | 5 | MDPI | SSCI, SCIE |

* SSCI—Social Sciences Citation Index; SCIE—Science Citation Index Expanded; ESCI—Emerging Sources Citation Index.

The first observation in the table is that most journals in the ranking were indexed in the Social Sciences Citation Index (SSCI), followed by two health-specific journals that were indexed in the Expanded Science Citation Index (SCIE) and two in the Emerging Sources Citation Index (ESCI), one being dedicated to the philosophy of management. Missing from this ranking are journals indexed in the Arts & Humanities Citation Index (AHCI) or works listed in the Book Citation Index (BCI).

We noticed that the papers were published in top journals dedicated to philosophy, respectively, in position three *Philosophy of Management*, and in position 13 is *Educational Philosophy and Theory*. Through this approach, philosophy is studied in terms of its potential to support and develop contemporary leadership, being considered as an "emerging social phenomenon" [58]. We highlight in this ranking the presence of at least six journals dedicated to management and anothertwo2 journals dedicated to leadership, which indicates the concern for studying and promoting this concept.

## 5. Discussion and Future Directions

From the literature review, we also identified papers containing bibliometric analyses, one of which was made over a long period of time (97 years), respectively, the period 1923–2019, and which reflects a concentration of 92% of the papers related to the concept of leadership analyzed in the last 30 years of the analysis [59]. From the representation given by the authors of the mentioned study, the period of the last 30 years had a different dynamic, in which the first 20 years of the period (1990–2009) reflected a publication interest of 26%, compared to the substantial increase manifested in the last 10 years (2010–2019) which represents 66% of the total published papers in the entire period of 97 years analyzed [59]. We believe that these results were influenced by the facilities offered by the Internet and indexing of scientific journals in open access databases, which encouraged the publication of studies and scientific papers on the concept of leadership in order to promote and implement it in the business area. Other research restricted the analyzed period, respectively, 1990–2018, limiting itself to the study of the concept of sustainable leadership, the authors identifying links with other essential concepts such as development, ethics and climate change [60], all highlighting the need to create and maintain interdisciplinary links leadership training. The approaches of other authors were made through the concept of sustainable leadership studied in connection with the field of management [61] or through their belonging to the contemporary concept of social responsibility [62], or through the orientation towards the future [63,64]. Sustainable leadership was conceptually analyzed to highlight its characteristics [65,66], but also in terms of papers published on this topic [67] continued by outlining future directions [68].

Compared to previous studies, the originality of our paper was given by the analyzed period 1979–2021, in which leadership showed a major evolution, and which was analyzed along with the contemporary trend of philosophical practices, which has already developed some interdisciplinary links to support the business area. We can consider this process an innovative one, which positions the philosophical practice within the process of development and implementation of the leader, and with the help of the concept of philosophical leadership, which can be promoted by current specialists in organizational philosophical consulting.

Next, we consider the questions expressed in the research hypothesis of the paper in order to present some conclusions that will contain the answers to them. Thus, we highlighted in the paper the tendencies to promote philosophical practices, which, since the 1980s, have been present in many countries around the world, by the fact that some philosophers have left university chairs and academia, migrating to the applied field of philosophy, currently manifested in the form of philosophical and applied ethics counseling. These tendencies toward philosophical practices have created specialists with the help of training programs started in universities and later by professional associations, interested in the continuous training of philosophical practitioners. Many of the specific tools used by practitioners and specialists in philosophical counseling can be implemented in personal

development and leadership programs, already tested in organizational consulting applications, where new specialists have entered. All these approaches and actions we believe will confirm the role that philosophical practice applied organizationally has in the process of developing contemporary leadership (Q1).

To answer the second question, we turn our attention to all the factors that participate in the leadership formation and development process, where we identify a first stage, presented schematically in Figure 5, and which refers to Business Education that usually benefits any manager.

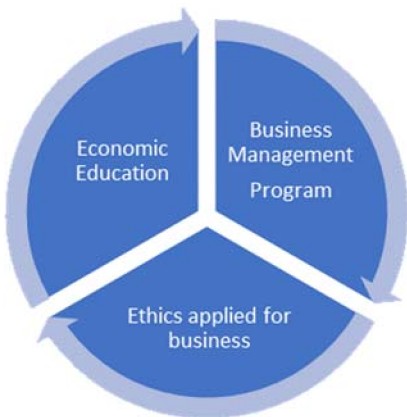

**Figure 5.** Business Education components involved in leadership.

We notice that the field of business training that benefits a manager or leader, contains, first, the basic economic education necessary for the specialization for the business environment, followed by management and business administration programs and continued through programs that initiate them in the field of applied ethics, including in business, all being essential components in preparing the leader. The next stage identified in leadership development is given by the coaching and personal development service to which any manager or leader in his training can turn, the service being offered by specialists in the field. The process of coaching and personal development applied in leadership has some essential objectives, reflected in Figure 6, precisely to emphasize the importance of the program in the training and formation of leaders, intended to support their activity, to achieve them.

| Coaching process | | |
|---|---|---|
| Leadership attitude | Company goals | Personal development |

**Figure 6.** The main objectives of coaching.

We observe, from Figure 6, which are the directions of action of a coach, oriented towards the development of the trained person in reaching some company goals, usually oriented towards achieving profit indicators as well as developing leadership skills by changing the attitude and behavior of the leader.

To improve the leadership process, we bring to your attention some tools and working methods existing in philosophical practice, and to which any specialist or practitioner in organizational philosophical counseling appeals. Speaking of the leader formation process,

we believe that these forms and means of applied practice can become part of a process called philosophical leadership, which can include them in building an integrated program for leaders and housekeepers working in organizations. We further reproduce a schematic structure of them, expressed in the following Figure 7.

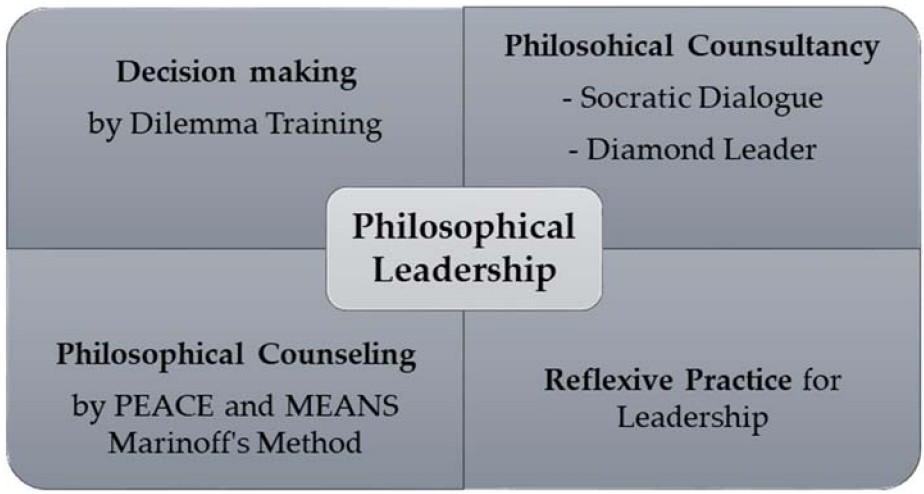

**Figure 7.** The main elements of philosophical leadership.

The schematic representation mainly refers to the main forms of action identified by practitioners, to which we attached tools, techniques or working methods used by specialists working in the organizational environment. We refer to the following directions outlined by practitioners as follows: decision making, achieved through Dilemma training or Philosophical consultancy by the Socratic dialog [10,36]) or the Diamond Leader method [34]; Philosophical counseling using the Marinoff's methods [37,38] or the Reflexive Practice for leadership development [23,24,26].

All these proposals for implementing the tools used in philosophical practices can be accessed by leaders and managers through a new specialization, organizational philosophical counseling, with a tendency toward professionalism, such as coaching services, already recognized and adopted by the business environment.

The analysis of published papers has highlighted various interdisciplinary links, which can connect philosophy to the concept of leadership, through various approaches, such as attracting wisdom [69] or interfering with engineering [70], approaches from existential [71] or ethical perspectives combined with those of personal development seen in global perspectives [72], or of communicational and managerial-type, adapted to the pandemic crisis [73,74]. Starting from the existing interdisciplinary links in the field of leadership training, I propose the realization of a complex training and personal development program for managers and leaders in particular, which we simply call the Leadership Training Program (LTP), whose origins can be represented in the following Figure.

From Figure 8, we observe that all three directions of action mentioned in the paper are brought together, in which the central role is played by the basic program of economic and business education, completed by coaching and personal development programs, but also by the proposed program, Philosophical leadership, which brings together the applied philosophical trend in the training format necessary to complete a leader. Thus, all these concerns can become essential components in promoting a unitary package of leadership training services, designed to train or improve a manager. To answer the last question of the paper (Q3) we turn to the following Figure 9, which shows us the evolutionary process of the leader training and formation programs, and which shows the sense of access but also the way of adapting them, in order to improve the leaders' services that can be provided by several specialists.

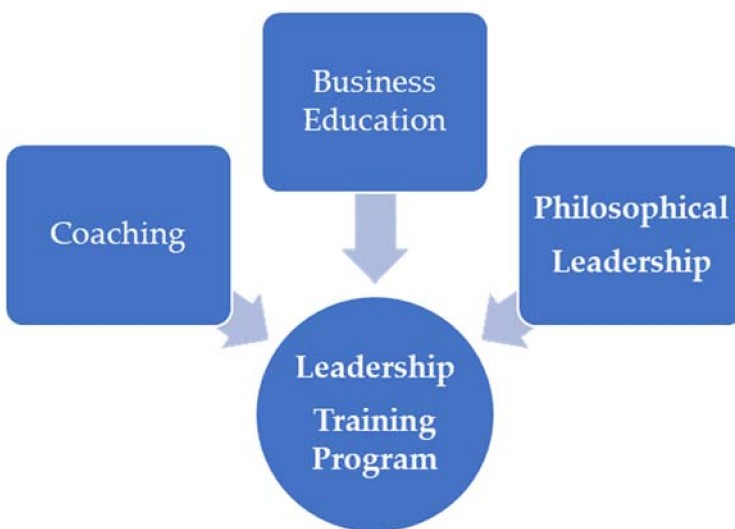

**Figure 8.** Components of the proposed leadership training program.

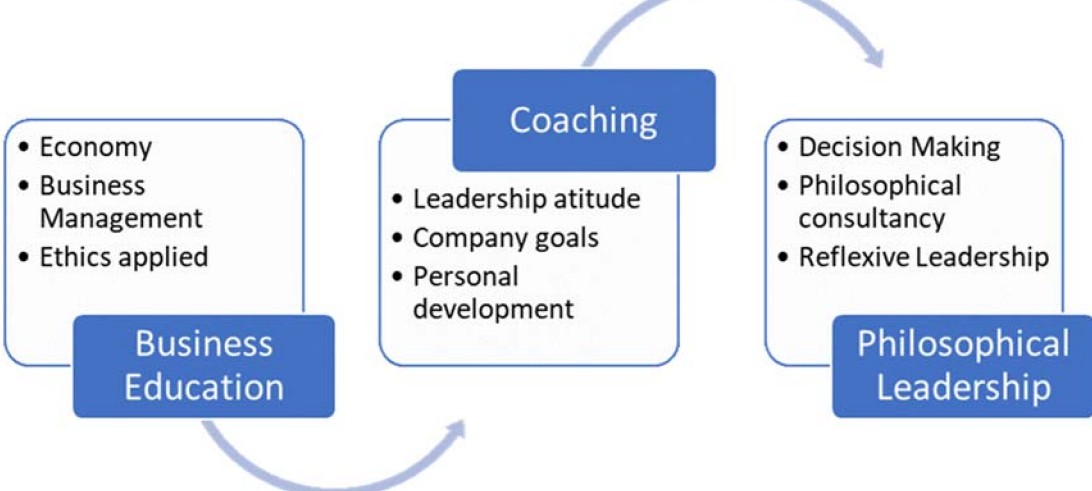

**Figure 9.** Areas involved in leadership training and education.

From the representation of the fields involved in leadership formation, we can deduce which are the main actors acting, where, on the one hand, is the manager or leader as beneficiaries of the program, along with the organization that supports and finances it in its training and development. However, there are the specialists we referred to in the paper, initially those from the educational system dedicated to business and management training, followed by specialists in coaching and personal development, who are joined by the proposed program and specialists in organizational philosophical consulting concerned with applying the concept called philosophical leadership. We can consider this process an innovative one, one that confirms the adaptation that can have the philosophical practice for organizations, which can use forms already dedicated in individual and group philosophical counseling, by developing and implementing in the format of any business training and philosophical leadership trend defined here.

Of course, there have been other approaches to leadership research, such as the psychological exploration (specific to the twentieth century), based on the common theory of the "great man", continued by the philosophical paradigm the "spirit of the time" of Marx and Engels, or the contingent model of leadership effectiveness [75], the theory of spiritual leadership, with an accent to sustainability [76,77].

The performed analysis had some limitations given by the sample of analyzed papers, taken from a single database (WoS), which led to the limitation of the identification of other papers published even in other languages, but the research can be continued by a future bibliometric analysis of articles indexed in other databases.

Of course, there are also other journals on the topic, which have not yet been indexed in WoS databases, and which have potential through the published papers, being written in various languages (English, Spanish, Italian, Korean, etc.) and promoted by the national associations of practitioners. It remains as a future direction, to expand the research to this aspect, insofar as the databases could be unified, to be introduced in the bibliometric analysis started within this topic.

## 6. Conclusions

The paper shows that leadership is focused on enhancing the human resources of an organization or community, whose training begins with the specific educational process, usually conducted in the form of management studies, followed by other forms of personal development or training, the type of coaching, in order to develop the skills necessary for a leader, including to stimulate his creativity. Thus, the transition from business management to the creation of the organizational leader occurs, and through the intervention we propose, to attract in this process the philosophical practice in preparing a leader, his personal development is stimulated, by facilitating the creative process, obtaining balance and developing a vision of a person's life. With the help of the tools offered by the philosophical practice, starting with the dialogical ones and continuing with the reflexive ones, the leader will develop new abilities which help him make the transition from leader to person through his own evolution on all levels, including professional [78]. Achieving this goal can be achieved by creating specialized programs and services in organizational consulting, based on the concept of philosophical leadership, in which practitioners and specialists in philosophical consulting can be involved in achieving these major objectives, and the beneficiaries can be the organization and the community in which they activate, but also the person who becomes a leader. The proposed innovation makes it possible for counseling specialists to actively participate in the leadership training program, as agents of change, for a sustainable development of organizations. We support the need to promote the new specialization of organizational philosophical consultant as well as the development of training and organizational consulting programs for managers and leaders to develop a specific profile of philosophical leadership, accumulated by specific skills in the field of philosophical practice.

In EU countries, there are concerns for supporting entrepreneurship as an element of innovation and sustainable business development [79], to which we believe that the philosophical tools specific to leadership that can be applied to entrepreneurs with a leadership vocation can also contribute successfully. We believe that the implications of philosophical leadership can be multiple, in the sense that philosophical practice successfully enters the business environment through specialized services for teams of leaders and managers, but also in the entrepreneurial field by applying the same concepts to those who develop new businesses [22].

Philosophical leadership has implications both in the personal development of leaders and in achieving the organization's goals, along with implementing the new ecological trend [80] by counseling to protect the environment in which organizations operate, but also in developing strategies specific to social responsibility that can be applied in the community.

**Author Contributions:** Conceptualization, V.-P.H. and C.-D.H.; methodology, V.-P.H. and C.-D.H.; software, C.-D.H.; validation, C.-D.H.; formal analysis, C.-D.H.; investigation, V.-P.H. and C.-D.H.; resources, V.-P.H.; data curation, C.-D.H.; writing—original draft preparation, V.-P.H. and C.-D.H.; writing—review and editing, V.-P.H. and C.-D.H.; visualization, V.-P.H. and C.-D.H.; supervision, V.-P.H.; project administration, V.-P.H.; funding acquisition, V.-P.H. and C.-D.H. All authors have read and agreed to the published version of the manuscript.

**Funding:** This article was funded through the FDI-2021-0414 project—"Developing the institutional capacity of the West University of Timișoara regarding excellence in scientific research".

**Institutional Review Board Statement:** Not applicable.

**Informed Consent Statement:** Not applicable.

**Data Availability Statement:** The data were collected from the following link: https://www-webofscience-com.am.e-nformation.ro/wos/woscc/summary/4ccf1a79-25ce-4cee-bb8e-95fddeffcd22-00fd125e/date-descending/1, (accessed on 20 June 2021).

**Conflicts of Interest:** The authors declare no conflict of interest.

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
