# Peer review of "Sustainable Leadership: Philosophical and Practical Approach in Organizations"

_sustainability, doi:10.3390/su13147918_

Round 1

Reviewer 1 Report

It is an article with a clear structure and coherence. However, it is an article that works on English-written articles (and some Italian books). Therefore, there is a lot of information missing in other languages such as Dutch, Spanish or German. Since the article is working on WoS-Clarivate and there is just one journal on that index in the world edited in Spain that includes English articles (HASER), it misses this information and the info of other journals such as APPA journal, Journal for Humanities Therapies (in Korea), Italian journals (Pragma, Phronesis) and the like. 

My proposal is to change slightly the title to locate the place of the research and, if possible, to include more information coming for other languages.

Author Response

Thank you for reviewing our paper and for your appreciation.

We took into account the proposed suggestion, to consider papers published in other languages, which is why we wrote this in the Discussions section, as one of the limits of this research, the paper taking in the bibliometric analysis only the papers identified in WOS (in English) which led to the impossibility of identifying other papers (published in other languages) that could be included in sample, which aspect can be considered as a limitation. An argument of this choice is given by the fact that although there are other specialized journals in the studied field, they have not indexed in WoS, and we refer to journals of practitioners from USA, Italy, South Korea (eg Philosophical Practice -APPA journal, Phronesis, Journal of Humanities Therapy). HASER journal which include papers in English, being indexed in ESCI-WoS, it was taken into account in the bibliometric analysis, but the results regarding the links between the studied concepts were less relevant due to the small number of citations (6) accumulated by the all indexed papers (61).

We indicate at the end of the paper among the future directions, that of expanding research in this regard, insofar as the databases could be unified, to be introduced in the bibliometric analysis, which is why in this paper we refer to this impediment and we show the limitations that emerged in our research.

For the same reasons we believe that it is not appropriate to change the title of the proposed paper, given that we stated in the abstract that the paper contains the analysis of works identified in WoS, motivation that we consider sufficient to place our research correctly, and avoiding a delimitation of the paper after the language used to carry out the works that were analyzed in the research.

Thank you for your interest, with the hope that you will accept our arguments, along with the improvement of the paper to be published in the journal Sustainability.

Reviewer 2 Report

The authors address a very important topic.  Ayman and Lauritsen note that the 20th-century psychological exploration of leadership research was initially based on the common philosophical perception of the “great man”. Later the exploration of leadership turn to situational theory which was based on Marx and Engels’s (Zeitgeist or “spirit of the time”) philosophical paradigm [Roya Ayman and Matthew Lauritsen. "Contingencies, Context, Situation, and Leadership," in John Antonakis and David V. Day (eds). The Nature of Leadership. 3rd ed. Thousand Oaks, CA: Sage, 2018].  

It seem that attention to leadership scholars like Jody Fry (esp. his model of spiritual leadership and emphasis on sustainability) could enhance this study. 

I also note that Leadership and the Humanities (a peer-reviewed international journal) has not been included in this study.  

Author Response

Thank you for the careful review of the paper and for the recommendations made. We have included in the comments in the results section three more significant references that improve the content, showing other approaches to researching the concept of leadership (eg. Ayman and Fry).

Regarding the indicated journal (Leadership and the Humanities), this was taken into account in the bibliometric analysis, but the results obtained were not relevant due to the small number of published articles (35) and the citations accumulated by them (10). We added in the Results section a comment on the limitations we adhered to, by choosing the sample for bibliometric analysis, also identifying as a future direction the potential to expand the research on other databases with interest in the field.

Thank you for your interest, with the hope that you will accept our arguments, along with the improvement of the paper to be published in the journal Sustainability.

Reviewer 3 Report

Thank you for the opportunity given in reviewing this interesting and current article. The author clearly state what the main purpose or objective of this article is so that readers can quickly identify!

Also, the authors does briefly mention the results and conclusions of his study.
In my opinion, some suggestions  regarding conclusions, are welcome to improve the quality of information presentation to readers as follows:

The conclusions are a bit short and a little evasive. I suggest the authors to highlight more the implications it has on the researched field!

Author Response

Thank you for the review of the paper and for the recommendations made. We have included in the conclusion more arguments to highlight the implications on the researched field.
We hope that you will accept new version of the conclusions along with all improvement of the paper, and you agree the paper to be published in the journal Sustainability.